The impact of nurses’ stress situation coping on somatization: a mediated moderation model

Qi Xiaoyan 1 ellaea2134392@163.com
Xu Hong-Ning 2
1 School of Nursing, Anhui Medical University , Hefei, Anhui , China
2 Pediatric Department, Anhui Children’s Hospital , Hefei, Anhui , China
Zhang Xin
Electronic publication date: 2024 Dec 6
Publication date: 2024
Volume: 12
Electronic Location ID: e18658
Received 2024 Sep 4; Accepted 2024 Nov 17
Copyright: © 2024 Qi and Xu
Copyright year: 2024
Copyright holder: Qi and Xu
License: This is an open access article distributed under the terms of the Creative Commons Attribution License, which permits unrestricted use, distribution, reproduction and adaptation in any medium and for any purpose provided that it is properly attributed. For attribution, the original author(s), title, publication source (PeerJ) and either DOI or URL of the article must be cited.
License URL: https://creativecommons.org/licenses/by/4.0/

Keywords: Nurses, Stress situation coping, Perceived social support, Depression, Somatization symptoms

Funding: Youth Science Fund of Anhui Medical University 2022xkj014 Social Science Fund of Anhui Provincial Department of Education 2022AH050635 Our study is supported by the Youth Science Fund of Anhui Medical University (2022xkj014) and the Social Science Fund of Anhui Provincial Department of Education (2022AH050635). The funders had no role in study design, data collection and analysis, decision to publish, or preparation of the manuscript.

==============================
Aims

The purpose of this study is to investigate the impact of nurses’ stress situation coping levels on somatization symptoms, the mediating effect of depression, and the moderating effect of perceived social support.

Background

As a core part of the global healthcare system, nurses are crucial to patient health and rehabilitation outcomes. However, due to heavy workloads, unreasonable staffing structures, and constant exposure to occupational risks in their workplaces, these factors often place nurses in a high-intensity, high-stress environment, which directly affects their physical and mental health and professional burnout. Coping with stress situations is an important means to help individuals effectively manage stress, reduce negative impacts, and maintain mental health. However, research on the impact of nurses’ stress situation coping methods on somatization symptoms is limited.

Methods

This study conducted a cross-sectional survey from December 2022 to April 2023, and finally included 293 nurses working on the front line of tertiary hospitals. Data was collected through questionnaires and analyzed using SPSS 24.0 and the SPSS macro program’s Models 7 and 14. This study is an observational study, strictly prepared and presented according to the STROBE checklist.

Results

The somatization symptom scores of the nurses were (27.27 ± 7.33) points, the stress situation coping scores were (59.90 ± 9.64) points, the perceived social support scores were (65.64 ± 12.90) points, and the depression scores were (4.42 ± 4.12) points. The somatization symptom scores of the nurses were positively correlated with the stress situation coping scores and depression scores, and negatively correlated with the perceived social support scores, with correlation coefficients of 0.200, 0.851, and −0.302, respectively. The stress coping level of the nurses had a direct positive impact on somatization symptoms (β = 0.081, p = 0.009), and a direct positive impact on nurse depression (β = 0.202, p = 0.001). Depression had a significant positive impact on somatization symptoms (β = 0.823, p = 0.000), and depression played a mediating role. Perceived social support had a significant negative impact on depression (β = −0.383, p = 0.000). Perceived social support had a significant moderating effect on the relationship between the stress situation coping level and depression (β = −0.121, p = 0.003).

Conclusion

In this study, the stress situation coping methods of nurses had a positive impact on somatization symptoms, had a complete mediating effect on the depression level of nurses, and perceived social support had a significant moderating effect in the pathway from the stress situation coping methods of nurses to depression-mediated somatization symptoms.

Introduction

Somatization symptoms are a complex set of clinical phenomena that are often characterized by a variety of changing somatic symptoms that typically lack a clear organic basis. Patients may have emotional problems or psychological disorders, but these emotional issues are converted into the external manifestation of somatic symptoms (Fu et al., 2019). The clinical significance of somatization symptoms lies in the fact that they may be a physical expression of psychological distress, which may be related to an individual’s personality traits, life experiences, social environment, and ways of coping with stress.

Somatic Symptom Disorder (SSD) is a chronic mental disorder characterized by excessive attention to and worry about somatic symptoms that are disproportionate to the actual severity of the symptoms and result in impairment of the patient’s daily functioning. After the COVID-19 pandemic, many recovered patients still have sequelae symptoms in somatic and mental systems within 12 months, which has made somatization symptoms a hot topic in academia again (Zeng et al., 2023). These symptoms not only affect the quality of life of patients but also impose an additional burden on medical resources. The evaluation of psychological variables through validated instruments is crucial to ensure the accuracy of results, as demonstrated by validation studies of psychometric scales (Diotaiuti, Valente & Mancone, 2021).

In the nursing industry, nurses are more prone to somatization symptoms due to the special nature of their work, such as shift work, high-intensity work, and mental stress. Studies have shown that job burnout among nurses is related to somatization symptoms, which include sleep disorders and socio-psychological disabilities, severely affecting the quality of life and sense of well-being of nurses (Sperling et al., 2023). Therefore, this study aims to explore the specific factors related to somatization symptoms in nurses, in hopes of finding effective ways to alleviate these symptoms and enhance their sense of well-being. The goal is to provide deeper insights into the nursing industry and to provide a scientific basis for the development of effective intervention strategies.

Coping with stressful situations refers to the cognitive and behavioral strategies that individuals adopt when facing stress or challenges (Marques et al., 2011; McWilliams, Cox & Enns, 2003). Coping can be the psychological and behavioral processes that individuals undertake to manage external or internal demands and to reduce stress-related discomfort. It includes problem-focused coping as well as emotion-focused coping. Compared with other professions, coupled with the inherent characteristics of nursing work, such as the heavy workload caused by the extreme imbalance of nurse-to-patient ratios and the occupational risk stress due to frequent contact with patients, nurses often exhibit greater work pressure and occupational risk stress (Chesak et al., 2019; Velana & Rinkenauer, 2021). The reasons may be that nurses worry about errors and accidents at work, fewer opportunities for further education, excessive time spent on writing medical records, uncooperative patients, and the low social status of nursing work, requiring targeted stress coping management (Jiang, Fan & Xue, 2024). In China, compared to nursing staff in other countries, the stress level of nurses is generally higher, which means they face severe psychological distress and are further prone to burnout and post-traumatic stress disorder (Ramachandran et al., 2023). The relationship between the physical and mental health of nurses and their level of coping with stressful situations has been confirmed (Yue et al., 2021). Active coping strategies, such as problem-solving and seeking social support, may reduce stress and decrease the occurrence of somatization symptoms. In contrast, passive coping strategies, such as avoidance, denial, and self-blame, may not effectively address the issues at hand and could even exacerbate stress and somatization symptoms, leading to compassion fatigue and severely affecting the physical and mental health of nurses.

The impact of different types of coping strategies on somatization and depressive symptoms is multifaceted. However, there is little research on how the way nurses cope with stressful situations affects their somatization symptoms. Therefore, this study investigates how coping with stressful situations affects the somatization symptoms of nurses and explores the specific associations between them.

Depression, as a common mental illness, severely affects individuals’ social functioning and quality of life. It is estimated that depression affects more than 350 million people. Globally, it is estimated that one in every 20 people reports having suffered from depression in the past year. Relevant studies have shown that nurses, working in high-stress and demanding environments, often suffer from depression, and the likelihood of registered nurses suffering from depression is almost twice as high as in other professions (Chen & Meier, 2021; Brandford & Reed, 2016). Data from a study in China found that the overall prevalence of depressive symptoms among Chinese nurses is 43.83%, much higher than in other countries (Xie et al., 2020). Depression affects nurses’ work capacity, work performance, interpersonal relationships, physical and mental health, and quality of life. In severe cases, it can lead to patient suicide. Nurses with a tendency towards depression are more likely to exhibit lower levels of stress coping when working, such as low mood, indifference, lack of motivation, and decreased self-confidence (Xuqing et al., 2013). When nurses exhibit negative stress coping in the workplace (such as professional burnout), their levels of depression are also generally high. Researchers have also proposed that by paying attention to the issue of nurse burnout, the level of depression among nursing staff can be effectively reduced (Chen & Meier, 2021).

Nurses, due to heavy workloads and work stress, are prone to depression. Fatigue, low mood, and despair are not only common main symptoms of the depressed group but also negative coping methods of nurses when facing job stress (Fan et al., 2023). When nurses have lower levels of depression, they exhibit a more positive work state, such as strong work engagement and high enthusiasm (Ran et al., 2020). Previous studies have mainly focused on the levels of somatization symptoms and depression in individuals or patients, and there is less research on the specific correlation between the two. In particular, no studies have been found on the relationship between depression and somatization symptoms in nurses. As the main force of the healthcare industry, studying the relationship between nurse depression and somatization symptoms, as well as exploring possible influencing factors, is crucial. Therefore, this study uses the level of depression as an intermediary variable between stress situation coping methods and somatization symptoms.

Perceived social support involves how individuals perceive and understand the help and resources they receive from social relationships. Social support can come from various channels, including family, friends, colleagues, communities, as well as government and non-government organizations (Xiao & Yang, 1987). Data from outpatient visits of psychiatric patients found that good perceived social support can improve a person’s well-being and also affect the treatment outcomes of somatoform disorders. When individuals receive support from the outside world, their somatization symptoms are also less apparent (Gerges, Hallit & Hallit, 2023). However, there is little research on somatization symptoms and social support in the nursing population. Therefore, this study attempts to use perceived social support as a moderating variable of stress situation coping and depression, trying to confirm the specific relationship between depression as a mediating variable and perceived social support as a moderating variable between stress situation coping methods and somatization symptoms.

Based on the above analysis, we have made the following hypotheses (as shown in Fig. 1):

Hypothesis 1: The way nurses cope with stress situations affects their somatization symptoms.

Hypothesis 2: The level of depression in nurses serves as a mediating factor between stress situation coping methods and somatization symptoms.

Hypothesis 3: Perceived social support moderates the early part of the mediating process between nurses’ stress situation coping methods and somatization symptoms, that is, through the level of depression.

Figure 1 Mediating and moderating effect model.

Methodology

Research design

This study is a cross-sectional survey that tests a hypothetical model to examine the mediating effect of depressive levels and the moderating effect of perceived social support on the relationship between the stress situation coping levels and somatization symptoms among nurses. A cross-sectional survey research design can efficiently and rapidly collect data, reflecting the mental health status of nurses at a specific point in time, especially the relationships between their stress-coping strategies, depressive symptoms, perceived social support, and somatization symptoms. This design is suitable for quickly assessing the current status of these variables among the nursing population, providing a foundation for further exploration of potential psychological interventions and possible future longitudinal studies. Through one-time data collection, the study can describe the prevalence and related factors among the nursing population, while also assessing the mediating role of depressive symptoms and the moderating role of perceived social support. This is crucial for developing targeted psychological support and intervention measures to improve the mental health and job performance of nurses.

Research sample

We chose three large tertiary hospitals in Hefei, the capital city of Anhui Province, as the research sites. These hospitals were selected due to their large size, high patient volume, and substantial nursing workload, ensuring the representativeness of the sample. They have extensive experience in providing nursing services, covering various types of nursing work and patient groups. The geographical location of these hospitals was also considered to facilitate on-site visits and data collection by the research team using a convenient sampling method, selecting three representative large tertiary hospitals in Hefei. Convenience sampling is a non-probability sampling method that relies on the convenience of the researcher to select participants. In this study, we selected three large tertiary hospitals in Hefei because they have cooperative relationships with our research institution and can provide the necessary support and resources. Moreover, these hospitals have a large number of nurses, which facilitates the collection of sufficient data. Although convenience sampling may not be as representative as random sampling, it is a practical choice when resources are limited. The inclusion criteria for participants were: (1) Clinical nurses with a work duration of no less than one year to ensure they have sufficient nursing experience and a deep understanding of nursing work. (2) Full-time nurses capable of independently undertaking the daily nursing work affairs of their departments to ensure they have sufficient responsibility and autonomy in their work; the exclusion criteria were: (1) Nurses who have not yet stabilized into rotating positions, as their work environment and responsibilities may still be changing, which could affect the research results. (2) Nurses on maternity leave, sick leave, or personal leave, as they may not be able to provide continuous work data, or their conditions may be irrelevant to the research topic. (3) Nurses studying abroad during the survey period, as they may not be able to participate in the entire research process, or their experiences may differ from those of domestic nurses. (4) Nurses unwilling to cooperate with researchers, as the validity of the research depends on the cooperation and honest responses of the participants.

We used anonymous online questionnaires to collect data to protect the privacy of the participants and increase the response rate. The questionnaire was designed through the “Questionnaire Star” platform and sent to the nursing group via WeChat groups, inviting them to fill in the questionnaire online. This method not only facilitates the collection and management of data but also ensures the anonymity and security of the data. The survey attracted the participation of 324 nurses. After screening, a total of 293 questionnaires met the requirements, with an effective recovery rate of 90.4%. This indicates that our sample has a high level of participation and cooperation. For the data that could not be collected due to the withdrawal of some participants from the study, this study employed sensitivity analysis to address these missing values.

All nurses who participated in the survey did so voluntarily and had clearly expressed their consent. It should be particularly noted that none of the nurses participating in this survey had religious beliefs, which helps us control for the variable of religious belief that may affect somatization symptoms.

Ethical approval

This study has been approved by the hospital’s ethics committee and strictly adheres to the guiding principles of the Declaration of Helsinki. Ethics Committee of Anhui Provincial Children’s Hospital approval to carry out the study within its facilities (Ethical Application Ref: EYLL-2021-018). Before the participants officially joined the study, we obtained their written informed consent. We are committed to safeguarding the rights and interests of all participants, ensuring that their rights are fully respected and protected. Furthermore, we clearly informed the participants that they would not receive any form of financial compensation for participating in this study, and we would not impose any financial burden on the participants during the data collection process. In this study, we have taken multiple measures to protect the confidentiality and security of participants’ data. First, we anonymized all collected data by removing any information that could identify individuals and replaced it with unique identifiers. Second, we provided training on data protection and privacy awareness to all team members. Finally, we ensured that the research complies with all applicable data protection regulations and signed data use agreements with participants, clarifying the purpose of data use and protective measures.

Measurement

In this study, the assessment scales for nurses include: a general information survey form, the Coping Inventory for Stressful Situations (CISS-21), the Perceived Social Support Scale, the Patient Health Questionnaire-9 (PHQ-9) depression scale, and the Self-Rating Scale for Somatization Symptoms (SSS). The use of validated scales, as highlighted by Diotaiuti, Valente & Mancone (2021), contributes to the robustness of the collected data and the convergence of the obtained results.

General information survey form

The general information survey form includes items on the nurses’ age, gender, years of work experience, professional title, place of residence, and history of taking psychiatric medication.

Self-rating scale for somatization symptoms

The Self-Rating Scale for Somatization Symptoms (SSS), after being introduced to China by Chinese scholars Zhuang et al. (2010), has been modified and consists of four parts: somatization symptoms, anxiety symptoms, depressive symptoms, and a combined anxiety-depressive symptoms section, with a total of 20 items. The scale uses a four-point scoring system: 1 point represents none (the symptom has not occurred during discomfort or illness), 2 points represent mild (the symptom is present during discomfort or illness but does not affect daily life), 3 points represent moderate (the symptom is present during discomfort or illness and the individual wishes to receive treatment or alleviation), and 4 points represent severe (the symptom is present during discomfort or illness and significantly affects daily life). Participants self-rate based on their actual situation, and the scores of all items are summed to get the total score. According to the scoring results, an SSS total score of less than 30 points indicates a basically normal psychological state with no obvious emotional issues; a total score of 30 points or more suggests the possible presence of psychological emotional issues; 38 points or more suggests the possible presence of mild to moderate emotional issues; and 42 points or more indicates the possible presence of severe psychological emotional issues. The Somatization Symptom Scale has been validated in various populations, including Asian populations. For instance, a study conducted among Chinese nursing staff also demonstrated its reliability and validity (Jin & Pan, 2016). In this study, the Chinese version of the SSS has an internal consistency Cronbach’s α coefficient of 0.836, with the Cronbach’s α coefficients for the factors ranging from 0.713 to 0.899; the split-half reliability of the SSS is 0.899, with the split-half reliability for the factors ranging from 0.681 to 0.876.

Perceived social support scale-12 items

The Perceived Social Support Scale, developed by Zimet et al. (1990) and revised by Jiang (1999), is used to assess the level of perceived social support among participants. It includes three dimensions: family support, friend support, and other support, with a total of 12 items. The scale uses a 7-point scoring system, with total scores ranging from 12 to 36 indicating a low support status, 37 to 60 indicating a moderate support status, and 61 to 84 indicating a high support status. The higher the total score of the scale, the more social support the individual feels they receive. Zhao et al. (2022) demonstrated that the Perceived Social Support Scale (PSSS) is effective and reliable for medical staff during public health emergencies. The combined explanatory variance of the three dimensions of the PSSS scale is 73.078%, and the correlation coefficients between the subscales range from 0.561 to 0.750, indicating a high structural validity of the PSSS scale. The Cronbach’s α coefficients for the total scale and the three subscales are 0.923, 0.909, 0.866, and 0.789, respectively, and their split-half reliability coefficients range from 0.801 to 0.915, indicating excellent reliability of the scale. In this study, the Cronbach’s α coefficient of this scale is 0.945.

PHQ-9 depression scale

The Chinese version of the PHQ-9 (Wang et al., 2014) consists of nine rating items. Patients rate the frequency of their symptoms based on their actual situation, with “not at all” scored as 0 points, “several days” as 1 point, “more than half the days” as 2 points, and “nearly every day” as 3 points. The sum of the scores of all items is the total score of the scale. A score of less than five indicates no depression, a score of five or more indicates possible mild depression, 10 or more indicates possible moderate depression, and 15 or more indicates possible severe depression. The Chinese version of the PHQ-9 has shown good reliability and validity in the Chinese population, making it suitable for depression assessment in various populations. In a study conducted by Sun et al. (2022), the Chinese version of the PHQ-9 used among nursing staff demonstrated good internal consistency, with a Cronbach’s α coefficient of 0.945.

Coping inventory for stressful situations

The Coping Inventory for Stressful Situations (CISS-21) (Li et al., 2017) is designed to measure the different coping strategies that people adopt when dealing with stressful situations. These coping strategies include three categories: task orientation, emotion orientation, and avoidance orientation. Task orientation refers to focusing on dealing with the current problem (such as arranging time reasonably, analyzing problems, etc.), emotion orientation refers to focusing on expressing emotions (such as worrying, self-blame, anger, etc.), and avoidance orientation refers to escaping from the problem to avoid distress (such as watching TV). Avoidance orientation can be further divided into disengagement and diversion. The questionnaire consists of 48 items, scored on a 5-point scale, with 16 items for each of the three coping tendencies, respectively. Higher scores may indicate that individuals tend to reduce the discomfort caused by stress through emotional regulation or avoidance strategies. As found in Zhai et al. (2021), the study showed that the application of the Chinese version of the CISS-21 in the nursing population shows that the scale has good internal consistency, with a Cronbach’s α coefficient of 0.945, indicating that the scale has a very high reliability. The Cronbach’s α coefficients for the three subscales are 0.88, 0.90, and 0.86.

Statistical methods

In this study, data analysis was conducted using IBM SPSS statistical software, version 24.0. The SPSS software is a powerful capabilities and widespread use as a statistical analysis tool capable of handling complex data analysis needs, including descriptive statistics, inferential statistics, and advanced statistical analyses. For categorical data, we used frequency counts and percentages for description, as these provide a clear representation of the distribution across different categories. For continuously distributed data that were normally distributed, we used the mean ± standard deviation for description, as these statistics offer information about the central tendency and dispersion of the data set. When comparing two groups of continuously distributed data that were normally distributed, we employed the Students’ t-test, a commonly used statistical test to determine if the mean differences between two independent samples are statistically significant. To analyze the relationship between different continuous variables, we utilized Pearson correlation analysis. This method is suitable for assessing the strength and direction of the linear relationship between two continuous variables.

The reason for selecting Model 7 and Model 14 macros in SPSS to test for moderated mediation effects is that these models offer a flexible and powerful way to evaluate complex relationships between variables. Model 7 is suitable for testing the impact of a moderating variable on the mediating effect, while Model 14 is used for testing the impact of a moderating variable on the relationship between the mediator and the dependent variable. These models employed the bootstrap method for bias correction, a resampling technique used to estimate the sampling distribution of a statistic without assuming a data distribution. We set a 95% confidence interval, a commonly used confidence level for estimating the credibility range of population parameters. The number of resamples was set to 5,000, a common sample size in the bootstrap method that provides stable and reliable estimation results. The significance level α was set at 0.05, a standard commonly used in social science research to control the probability of Type I errors.

Results

Comparison of somatization symptom scores among clinical nurses with different demographic characteristics

Among the 293 clinical nurses, there were two (0.7%) males and 291 (99.3%) females; their ages ranged from 24 to 58 (mean 33.69 ± 6.53) years old. The years of work experience ranged from 1 to 40 (mean 11.81 ± 7.34) years. For a detailed description of other demographic characteristics, see Table 1. Nurses who had previously taken psychotropic medications had higher total somatization symptom scores than those who had not, and the difference was statistically significant (p < 0.05), as detailed in Table 1. Additionally, the correlation between the age and years of work experience of the 293 clinical nurses with the total somatization symptom score was not significant (r = −0.033, 0.003, p = 0.575, 0.964).

Table 1 Comparison of somatization symptom scores among clinical nurses with different demographic characteristics.

Variable	n (%)	Somatization symptom scores, Mean ± SD	Statistic	p	
Total	293 (100)	27.27 ± 7.33			
Gender			t = −1.02	0.308	
Male	2 (0.68)	22.00 ± 2.83			
Female	291 (99.32)	27.31 ± 7.34			
Professional title			t = −0.24	0.810	
Entry-level	132 (45.05)	27.16 ± 7.82			
Intermediate and above	161 (54.95)	27.37 ± 6.93			
Place of residence			t = −0.49	0.627	
Rural or township	4 (1.37)	25.50 ± 5.45			
Urban	289 (98.63)	27.30 ± 7.36			
History of psychotropic medication use			t = −4.66	<0.001	
No	284 (96.93)	26.93 ± 6.98			
Yes	9 (3.07)	38.11 ± 10.13			
Note:

SD, standard deviation; t, t-test. p < 0.001 (in bold) usually indicates that the results have a high level of statistical significance.

Control and examination of common method bias

Since this study employed a questionnaire survey method, there might be an issue with common method bias. The Harman single-factor model method was used to test for common method bias. The results showed that there were three factors with eigenvalues greater than one, with the first factor accounting for 24.77% of the variance, which is less than the critical value of 40%. Therefore, there is no severe common method bias in this study.

Scoring of different scales among clinical nurses and correlation analysis

The somatization symptom score of the 293 clinical nurses was (27.27 ± 7.33) points, the depression score was (4.42 ± 4.12) points, the stress situation coping score was (59.90 ± 9.64) points, and the perceived social support score was (65.54 ± 12.90) points. For detailed scoring results of the specific scales, see Table 2. The correlation analysis results showed that the somatization symptom scores of the 293 clinical nurses were positively correlated with the stress situation coping scores and depression scores (p < 0.05), and negatively correlated with the perceived social support scores (p < 0.05). For the correlation analysis of other scale scores, see Table 2.

Table 2 Descriptive statistics and correlation analysis of scale scores for clinical nurses.

Variables	Mean	Standard deviation	1	2	3	4	
1 Coping score for stressful situations	59.90	9.64	1.00				
2 Perceived social support score	65.64	12.90	0.299**	1.00			
3 Depression score	4.42	4.12	0.142*	−0.292**	1.00		
4 Somatization symptom score	27.27	7.33	0.200**	−0.302**	0.851**	1.00	
Note:

Two (**) and one (*) asterisk represent the significance levels of 5% and 10%, respectively.

Mediation analysis

The mediating effect of perceived social support in the relationship between coping with stressful situations and depression

First, the four variables of perceived social support, depression, coping with stressful situations, and somatization symptoms were standardized. Then, with somatization symptoms as the dependent variable, coping with stressful situations as the independent variable, depression as the mediating variable, perceived social support as the moderating variable, and gender, age, years of work experience, professional title, place of residence, and history of psychotropic medication use as covariates, the Model 7 and Model 14 in the SPSS macro program were used to test the moderated mediation effect. The results showed that Model 7 was established (the first half of the mediation model is moderated), and it was found that nurses’ coping with stressful situations had a positive but non-significant effect on somatization symptoms (β = 0.081, p = 0.009); had a positive but non-significant effect on the level of depression in nurses (β = 0.202, p = 0.001); and the level of depression in nurses had a positive but non-significant effect on somatization symptoms (β = 0.823, p = 0.000); therefore, this model proves that the level of depression in nurses fully mediates the relationship between nurses’ coping with stressful situations and somatization symptoms (Fig. 2, Table 3).

Figure 2 Moderated mediation model diagram.

Table 3 Test of the moderating effect of perceived social support on the mediation model.

	Somatization symptoms (Dependent variable)	Depression (Dependent variable)	
	β	SE	t	p	β	SE	t	p value	
Constant	0.145	0.975	0.148	0.882	−0.398	1.772	−0.225	0.822	
Coping with stressful situations	0.081	0.031	2.634	0.009**	0.202	0.057	3.513	0.001**	
Perceived social support					−0.383	0.059	−6.53	0.000**	
Interaction of coping with stressful * Situations and perceived social support					−0.121	0.041	−2.974	0.003**	
Gender (Female vs. Male)	0.086	0.367	0.235	0.814	0.363	0.664	0.546	0.585	
Age	−0.032	0.018	−1.752	0.081	−0.037	0.032	−1.147	0.252	
Professional title (Intermediate and above vs. Entry-level)	−0.118	0.074	−1.587	0.114	0.164	0.131	1.251	0.212	
Years of work experience	0.037	0.016	2.355	0.019*	0.018	0.028	0.641	0.522	
Place of residence (Urban vs. Rural or township)	0.243	0.26	0.935	0.35	0.236	0.461	0.512	0.609	
History of psychotropic medication use	0.469	0.179	2.621	0.009**	0.86	0.315	2.733	0.007**	
Depression	0.823	0.031	26.406	0.000**					
Sample size	293	293	
R 2	0.746	0.204	
Adjusted R2	0.738	0.176	
F	F = 104.089, p = 0.000**	F = 8.053, p = 0.000**	
Note:

**, * represent the significance levels of 5% and 10%, respectively.

The moderating role of perceived social support in the resilience pathway from nurses’ coping strategies for stressful situations to depressive levels in nurses

The influence of perceived social support on nurses’ depression is negatively oriented but not significant (β = −0.383, p = 0.000), and the interaction term between coping with stressful situations and perceived social support significantly predicts depression (β = −0.121, p = 0.003), indicating that perceived social support plays a moderating role in the prediction of depression by coping with stressful situations. The results are shown in Table 3 and Fig. 1.

The results of the Conditional Indirect Effect indicate that for the mediating variable of depression, when the level of perceived social support is low, the Boot 95% CI does not include the number 0, meaning that there is a mediating effect at this level, and the coping strategies for stressful situations have a positive but non-significant impact on the level of depression. The Effect value is 0.266, with a 95% CI [0.142–0.388]; when the level of perceived social support is at the average level, the Boot 95% CI does not include the number 0. This means that there is a mediating effect at this level, indicating that the positive impact of coping strategies for stressful situations on depression is diminished, with an Effect value of 0.166, and a 95% CI [0.06–0.276], resulting in an Effect value decrease from 0.266 to 0.166; when the level of perceived social support is high, the Boot 95% CI includes the number 0, with an Effect value of 0.066 and a 95% CI [−0.046 to 0.201]. This means that there is no mediating effect at this level. In summary, the mediating effect is inconsistent at different levels of perceived social support (from low to medium), indicating a moderating mediating effect. The results can be seen in Table 4 and Fig. 3.

Table 4 Results of the conditional indirect effect.

Mediating variable	Perceived social support	Level value	Effect	BootLLCI	BootULCI	
Depression	Low level (Mean − 1SD)	−1	0.266	0.142	0.388	
Average value	0	0.166	0.06	0.276	
High level (Mean + 1SD)	1	0.066	−0.046	0.201	
Note:

BootLLCI refers to the lower limit of the 95% confidence interval obtained by Bootstrap resampling, BootULCI refers to the upper limit of the 95% confidence interval obtained by Bootstrap resampling, and the type of Bootstrap used is the bias-corrected Bootstrap method.

Figure 3 Analysis of the moderating effect of perceived social support between coping with stressful situations and depression.

Discussion

The validation of psychometric instruments, as noted by Diotaiuti, Valente & Mancone (2021), is essential for obtaining reliable and relevant data in psychological research, especially in contexts with high emotional variability such as nursing. In this study, we investigated how nurses’ somatization symptoms are affected by their coping strategies in stressful situations. Our theory regarding the high correlation between nurses’ coping with stressful situations, perceived social support, depression, and somatization symptoms was only partially confirmed.

Firstly, the average score for the level of depression among nurses in the study was (4.42 ± 4.12) points (Table 2). This score is above the average and indicates mild depression, a situation that has been confirmed in previous research by Chinese scholars (Zheng et al., 2021; Lei et al., 2022). The score for perceived social support among nurses was (65.54 ± 12.90) points, which is at a relatively high level overall. However, this is inconsistent with the results of the study by Xie et al. (2020), possibly because as China’s social health care system gradually improves, policy and organizational support for nurses has also increased. The study by Xie et al. (2020) took place during the peak of the outbreak, when the number of infected patients surged, and there was a lack of sufficient medical rescue teams and effective treatment methods, resulting in nurses’ level of social support being moderately low. Although some literature mentions the importance of social support for nurses, it does not specify the level of perceived social support that nurses experience. We found that the score for nurses’ somatization symptoms was (27.27 ± 7.33) points, indicating a moderately low level of somatization symptoms.

In this study, the score for nurses’ coping with stressful situations was (59.90 ± 9.64) points, indicating that nurses have certain strategies to face stress. This result is consistent with the findings of studies by Zhu et al. (2022) and Wang et al. (2022), possibly because nurses take certain measures to cope with workplace stress, professional pressure, and patient-doctor conflicts (Lin et al., 2023).

Secondly, the study confirms Hypothesis 1 that the way nurses cope with stressful situations affects somatization symptoms. The way nurses cope with stressful situations affects somatization symptoms. This is consistent with the conclusions of Betke, Basińska & Andruszkiewicz (2021) and Iwanowicz-Palus et al. (2022), and the possible reason is that nurses may adopt positive or negative coping strategies when dealing with stress (Xie, Li & Kuang, 2020). Positive coping strategies, such as problem-solving, emotional regulation, and seeking social support, may help to alleviate stress and reduce the occurrence of somatization symptoms. On the contrary, negative coping strategies, such as avoidance, denial, and self-blame, may not effectively solve problems and may even exacerbate stress and somatization symptoms. Even lead to compassion fatigue and autism spectrum disorders, which severely influence the physical and mental health of nurses (El-Ashry et al., 2023; Alruwaili et al., 2024a, 2024b; Abdelaziz et al., 2024). This study found that the way nurses cope with stressful situations is positively correlated with somatization symptoms. Mediating analysis indicates that the way of coping with stressful situations has a positive impact on nurses’ somatization symptoms, with a direct impact of β = 0.081, p < 0.009 (Table 3). The key factor causing this is the different ways nurses cope with stressful situations and the association with somatization symptoms (Chesak et al., 2019). The reason may be that the nurses in this study adopted more negative coping strategies for stressful situations, which can intensify negative emotions, reduce the ability to cope with stress, and increase stress perception. However, this relationship has not yet been further studied and confirmed. Different ways nurses cope with stressful situations, such astask-oriented coping strategies, which focus on solving problems, may reduce stress and lower somatization symptoms because they involve proactive actions and direct addressing of issues. Emotion-oriented coping strategies, such as expressing emotions and seeking emotional support, may alleviate depressive symptoms as they provide an outlet for emotional release and opportunities for social interaction. Whereas avoidance-oriented coping strategies, such as escaping and diverting attention, may provide short-term stress relief, but in the long run, they may increase somatization and depressive symptoms because they fail to address the root causes and may even intensify the individual’s escapist behaviors and emotional suppression. Individual and organizational strategies to reduce and prevent work stress, can control and alleviate the emotional state of nurses to a certain extent by avoiding or reducing work stress (Roberts & Grubb, 2014; Alzoubi et al., 2024), and the ventilation of these negative emotions can reduce the somatization symptoms of nurses (Ben-Ezra et al., 2013).

The study confirms Hypothesis 2 that the level of depression in nurses serves as a mediating factor between stress situation coping methods and somatization symptoms.

When nurses exhibit symptoms of depression, their somatization symptoms increase. According to the research results, the level of depression in nurses is positively correlated with somatization symptoms; the level of depression in nurses has a separate impact, and the severity of somatization symptoms in nurses worsens as the level of depression increases. The results of this study are consistent with the results of previous researchers (Chen et al., 2019), which found that the incidence of somatization symptoms in patients with depression was 65.0% to 98.2%, severely affecting the patients’ quality of life. According to the research results, the level of depression has a positive impact on the association between the way nurses cope with stressful situations and somatization symptoms (Table 3; Fig. 2). The way nurses cope with stressful situations increases the level of depression in patients, and this impact, in turn, leads to an increase in the level of somatization symptoms in nurses. Currently, the information on how the level of depression affects somatization symptoms is clear. However, according to relevant research findings (Li, Jia & Miao, 2019), somatic symptoms are significantly correlated with dietary conditions, major life events, and adverse drug reactions, and point out that somatic symptoms are positively correlated with the degree of depression. Further investigation is needed to verify the association from the perspective of this study and to attempt to propose that clinical medical staff should pay more attention to the somatization symptoms of patients with depression.

This study confirms Hypothesis 3 that theperceived social support, there is a moderating effect on the relationship between nurses’ levels of depression and their coping strategies for stressful situations. The results of this study show that the way nurses cope with stressful situations has a positive impact on their level of depression, regardless of the differences in the ways nurses cope with stressful situations. However, nurses with low levels of perceived social support experience significantly greater adverse effects than those with high levels of perceived social support (Effect value from 0.266 to 0.166). Similarly, as the level of perceived social support increases, the adverse impact of the nurses’ coping strategies for stressful situations on their level of depression decreases. A certain level of perceived social support serves as a buffer for the nurses’ coping strategies for stressful situations, providing additional support and assistance (Li, Da & Sun, 2010; Jin et al., 2015). When the level of perceived social support is high, both the nurses’ coping strategies for stressful situations and their level of depression are reduced, and when the level of perceived social support is low, both the nurses’ coping strategies for stressful situations and their level of depression increase. When the level of perceived social support is at a moderate level, there is no moderating effect between the nurses’ coping strategies for stressful situations and their level of depression. The possible reason for this is that the coping strategies adopted by nurses, such as active or passive coping, will affect their stress perception and mental health status. Active coping strategies help to alleviate stress, while passive coping may exacerbate stress and depressive emotions (Ying et al., 2019). It is suggested that by enhancing social support and cultivating coping strategies, improving the work environment, implementing regular mental health assessments, and adopting stress management methods and coping strategies, potential mental health issues among nurses can be identified in a timely manner, and corresponding intervention methods can be formulated to help nurses better cope with work pressure, improve their mental health levels, reduce the occurrence of depressive emotions, and indirectly reduce adverse somatization symptoms in nurses, hospital management can also provide support to nurses through positive informal communication, which can mitigate the negative effects of abuse and enhance team cohesion. Alternatively, by providing decent working conditions, such as reasonable working hours, fair compensation, and a good working environment, it can improve nurse job satisfaction, maintain the quality of care, and promote the overall well-being of the healthcare team (Zoromba et al., 2024; El-Gazar et al., 2024) thereby enhancing the physical and mental health and well-being of nurses.

Limitation and future research

This study, based on data from tertiary hospitals in China, accurately reflects how nursing staff respond to stress, their levels of depression, perceived social support, and somatization symptoms, but there are several limitations. Firstly, we utilized a convenience sampling method, and our sample was limited to nurses from tertiary hospitals in Hefei City, which may not represent a broader nursing population. Future studies should clearly discuss the limitations of generalizability and include a more diverse sample to verify findings in different contexts. This sampling method may lead to sample bias, affecting the generalizability of the results. Therefore, future research should consider using increase the number of sample and broader sampling methods, such as stratified random sampling, to enhance the representativeness of the sample. Second, because this study employed a cross-sectional design, it limits our ability to determine causal relationships between variables, so future studies could adopt a longitudinal design to better understand the dynamic relationships between variables. Additionally, since the data were collected through self-report questionnaires, this may introduce self-report bias, and it is recommended that future studies combine interviews, observations, and other objective methods to reduce such bias. We also recognize that variables such as personal lifestyle, work environment, the quality of social support received, variations in workload, shift patterns, and personal stressors outside of work, which may affect the psychological state of nurses and their ability to cope with stress, thereby indirectly affecting the manifestation of somatization symptoms, were not fully considered in this study and could impact the research outcomes. Thus, future studies should regard these factors as potential confounding variables.

Conclusion

The findings of this study reveal how perceived social support, acting as a moderating factor, impacts the somatization symptoms of frontline clinical nurses by influencing their stress-coping strategies and levels of depression. Particularly during the COVID-19 pandemic, the study further uncovers how nurses’ levels of depression, as a mediating variable, affect their stress-coping strategies and somatization symptoms. These discoveries have profound implications for practice and management in the field of nursing. Our results highlight the importance of considering the psychological health status of nurses in nursing management, which is crucial for enhancing job satisfaction and psychological well-being among nurses. By gaining an in-depth understanding of how nurses cope with work stress and how social support and depressive symptoms affect these coping strategies, we can provide scientific guidance for nursing practice, helping nurses manage work stress more effectively and thus maintain better physical and mental health.

It is recommended that hospitals and management teams implement a series of targeted interventions to alleviate the stress of nurses such as establishing social support networks, providing mental health education and resources: offering regular mental health education and easily accessible psychological services for nurses. Implement nurse work stress management programs, including stress relief workshops, flexible work arrangements, and adequate rest time, to alleviate work stress. Enhance coping strategies for nurses through training and education, such as cognitive-behavioral therapy and mindfulness meditation. Regularly monitor and assess the mental health status of nurses to identify issues promptly and provide necessary support. The data provided by this study lays the foundation for developing clinical intervention measures for adverse somatic symptoms in nursing personnel, which is essential for formulating effective therapeutic intervention plans.

Supplemental Information

Supplemental Information 1 Raw data.

Supplemental Information 2 English language codebook.

Supplemental Information 3 STROBE checklist.

Additional Information and Declarations

Competing Interests

Author Contributions

Human Ethics

Ethics

Data Availability

The authors declare that they have no competing interests

Xiaoyan Qi conceived and designed the experiments, performed the experiments, analyzed the data, prepared figures and/or tables, authored or reviewed drafts of the article, and approved the final draft.

Hong-Ning Xu conceived and designed the experiments, performed the experiments, prepared figures and/or tables, authored or reviewed drafts of the article, and approved the final draft.

The following information was supplied relating to ethical approvals (i.e., approving body and any reference numbers):

Ethics Committee of Anhui Provincial Children’s Hospital approval to carry out the study within its facilities (Ethical Application Ref: EYLL-2021-018).

The following information was supplied relating to ethical approvals (i.e., approving body and any reference numbers):

The Anhui Provincial Children’s Hospital granted ethical approval to carry out the study within its facilities (Ethical Application Ref: EYLL-2021-018).

The following information was supplied regarding data availability:

The raw data is available in the Supplemental File.

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
