# Peer review of "The impact of nurses’ stress situation coping on somatization: a mediated moderation model"

_PeerJ, doi:10.7717/peerj.18658_

## Round 0.1 · original submission · Major Revisions

The authors are requested to carefully revise the manuscript and answer the questions raised by the reviewers.

·

Basic reporting

I would like to express my deepest gratitude for the opportunity to review your work entitled "The impact of nurses’ Stress Situation Coping on Somatization: A mediated moderation model." Your dedication to this research provides a significant contribution to the field of mental health nursing, and I appreciate the meticulous approach you have taken in addressing such an important topic.

Importance of the Topic (Lines 1-7): The study highlights the importance of coping with stress among nurses and its impact on somatic symptoms, emphasizing the connection with depression and perceived social support.

Contribution to Nursing Practice (Lines 18-23): The authors effectively underscore how improving coping strategies and social support can positively impact nurses’ psychological health, providing valuable insights for optimizing nursing management.

Originality of the Research (Lines 25-41): The methodological structure and results clearly outline the mediating effect of depression and the moderating effect of social support, presenting a comprehensible and scientifically robust framework.

Suggestions for Revisions

Clarity of the Introduction (Lines 53-78): Enhance the introduction by specifying in more detail the research gaps this study aims to fill. Consider expanding on the description of somatization symptoms and their clinical implications.

Methodology Description (Lines 148-164): Provide a clearer explanation regarding the sample selection and the inclusion and exclusion criteria. For instance, include more details on how the hospitals were selected and justify the use of convenience sampling.

Statistical Analysis (Lines 240-249): Expand on the statistical analysis used for the moderated mediation model, explaining why SPSS Models 7 and 14 were chosen. Include a brief description of how data normalization was performed.

Conclusions (Lines 426-442): Rephrase the conclusions to more strongly emphasize the practical importance of the findings and suggest possible evidence-based interventions.

Sections to Remove or Revise

Redundancy in Results (Lines 324-327): Excessive repetition of the results across various sections makes the reading less fluid. I suggest reviewing these sections to avoid redundancy.

Limitations Not Adequately Explored (Lines 410-423): The authors do not clearly indicate the study’s limitations. I recommend including a more detailed discussion on how the sampling and potential unconsidered variables may have influenced the results.


Limited Sampling (Lines 417-423): The use of convenience sampling could introduce bias; it is advisable to mention how this limitation affects the results and their generalizability.
Sample Size (Lines 164-166): The relatively small number of participants might reduce the study's statistical power. This should be clarified, and expanding the population in future studies should be suggested.

The work presents a valuable and original contribution, but it would benefit from a review that emphasizes methodological clarity and more thoroughly addresses the study’s limitations to strengthen its rigor.

To further enrich your study and enhance the theoretical support regarding the analysis of the psychological variables used, I invite you to consider citing the following article:

Diotaiuti, P., Valente, G., & Mancone, S. (2021). Validation study of the Italian version of Temporal Focus Scale: psychometric properties and convergent validity. BMC Psychology, 9(1), 19. https://doi.org/10.1186/s40359-020-00510-5


Introduction (Lines 53-78): When describing the relevance of coping with stress and related psychological variables, you can integrate the citation to strengthen the theoretical basis regarding the psychometric validity of the instruments used for psychological assessment.

Suggested Placement:

“The evaluation of psychological variables through validated instruments is crucial to ensure the accuracy of results, as demonstrated by validation studies of psychometric scales (Diotaiuti et al., 2021).”

Methodology (Lines 179-183): When discussing the use of scales for measuring psychological variables, you can include the citation to support the reliability and validity of the psychological scales.

Suggested Placement:

“The use of validated scales, as highlighted by Diotaiuti et al. (2021), contributes to the robustness of the collected data and the convergence of the obtained results.”

Discussion (Lines 323-365): While arguing the importance of psychological measurements in understanding the dynamics of stress, depression, and social support, you can refer to the article to emphasize the significance of validating the instruments used.

Suggested Placement:

“The validation of psychometric instruments, as noted by Diotaiuti et al. (2021), is essential for obtaining reliable and relevant data in psychological research, especially in contexts with high emotional variability such as nursing.”

Experimental design

The study investigates the impact of nurses' stress-coping strategies on somatization symptoms, exploring the mediating role of depression and the moderating role of perceived social support. While the study addresses a significant topic, there are several areas where the experimental design falls short of the standards, along with suggested improvements:

Sampling Method: The study uses a convenience sampling method limited to tertiary hospitals in Hefei, which introduces potential sampling bias and limits the generalizability of the findings.

Suggested Improvement: Expand the sample to include diverse healthcare settings and use random sampling to reduce bias.
Study Design and Data Collection: The cross-sectional design limits the ability to infer causal relationships between the variables studied.

Suggested Improvement: Consider a longitudinal design in future studies to better assess causal links between stress-coping strategies, depression, and somatization symptoms.
Control Variables: The study does not adequately address potential confounding factors such as nurses’ workload, shift patterns, and personal life stressors that could influence coping strategies and somatization symptoms.

Suggested Improvement: Include a broader range of control variables that account for external stressors and personal characteristics.
Statistical Analysis: The use of SPSS Models 7 and 14 for moderated mediation is appropriate; however, the rationale for choosing these specific models is not well-explained, which may confuse readers unfamiliar with these methods.

Suggested Improvement: Provide a clearer explanation of why these models were chosen and discuss alternative statistical approaches that could be used.
Measurement Instruments: While the study uses validated scales, there is limited information on whether the scales were culturally adapted or validated in the specific context of Chinese nursing staff.

Suggested Improvement: Include details on any cultural adaptations of the scales and provide validity and reliability data specific to the study population.
Ethical Considerations: The study received ethical approval; however, there is no mention of how participant privacy was maintained beyond standard consent procedures, especially given the sensitive nature of mental health data.

Suggested Improvement: Clearly outline measures taken to protect participant confidentiality and data security.
Limitations Section: The limitations discussed are not sufficiently detailed. Important limitations, such as the potential for self-report bias in questionnaire responses, are not fully explored.

Suggested Improvement: Expand the limitations section to include discussion on self-report bias, the impact of the cross-sectional design, and the representativeness of the sample.
Overall Assessment: The experimental design of this study addresses an important topic but requires refinements to meet higher research standards. Implementing these suggestions will enhance the study's rigor and the credibility of its findings.

Validity of the findings

The study provides valuable insights into the relationship between nurses' stress-coping strategies, depression, and somatization symptoms. However, there are several areas where the validity of the findings could be strengthened:

Causal Inferences: The cross-sectional design limits the ability to establish causal relationships between the variables studied. The findings suggest associations rather than causations, which should be clearly stated to avoid misinterpretation.

Suggested Improvement: Explicitly acknowledge in the discussion that causality cannot be inferred due to the study design. Future studies could consider longitudinal designs to better explore causal pathways.
Control for Confounding Variables: The study does not sufficiently account for other potential confounders that could influence the observed relationships, such as variations in workload, shift patterns, and personal stressors outside of work.

Suggested Improvement: Incorporate a broader range of control variables in the analysis and discuss their potential impact on the findings. This would help strengthen the validity of the observed relationships.
Self-Report Bias: The data are collected using self-report questionnaires, which can introduce response biases, such as social desirability bias, particularly when assessing sensitive topics like depression and stress coping.

Suggested Improvement: Discuss the limitations of using self-report measures and consider employing objective measures or triangulating with additional data sources (e.g., clinical assessments or third-party observations) in future research.
Generalizability: The sample is limited to nurses from tertiary hospitals in Hefei, which may not be representative of the broader nursing population, particularly those working in different regions or settings.

Suggested Improvement: Clearly discuss the limits to generalizability and suggest that future studies include a more diverse sample to validate the findings across different contexts.
Statistical Analysis: The statistical models used (moderated mediation models) are appropriate; however, the interpretation of the findings should carefully address the assumptions of these models, including the robustness checks conducted.

Suggested Improvement: Include a section on how the assumptions of the statistical models were tested and any limitations identified in the analysis process.
Handling of Missing Data: The manuscript does not discuss how missing data were handled, which could impact the study's findings if not appropriately addressed.

Suggested Improvement: Provide details on the approach used to manage missing data, such as imputation methods or sensitivity analyses, to enhance the credibility of the results.
Overall Assessment: While the study offers valuable contributions, addressing these areas will significantly enhance the validity of the findings and provide a clearer understanding of the implications of the results. Your efforts to improve these aspects will help ensure that the conclusions drawn are robust and well-supported by the data.

·

Basic reporting

Positive Aspects:
Research Significance: The study addresses an important topic, as nurses often work in high-stress environments, and understanding the impact of stress coping strategies on somatization symptoms can lead to better mental health and job satisfaction for healthcare workers.
Litrature should mention the folowing articles; Mistreatment of nurses by patients and its impact on their caring behaviors: The roles of psychological detachment and supervisor positive gossip, https://doi.org/10.1111/inr.12970 and The role of psychological ownership in linking decent work to nurses' vigor at work: A two-wave study https://doi.org/10.1111/jnu.12970
Also, use this study "Alzoubi, M. M., Al-Mugheed, K., Oweidat, I., Alrahbeni, T., Alnaeem, M. M., Alabdullah, A. A. S., ... & Hendy, A. (2024). Moderating role of relationships between workloads, job burnout, turnover intention, and healthcare quality among nurses. BMC psychology, 12(1), 1-9."
Clear Hypothesis: The study's hypotheses, particularly regarding the mediating role of depression and moderating role of perceived social support, are well-justified and contribute to the existing literature on nursing stress and mental health.
Comprehensive Methodology: The study uses a robust methodology, including validated scales such as the PHQ-9 and Perceived Social Support Scale, which enhances the reliability of the findings.
Sample Size and Ethical Considerations: The sample size is adequate, and ethical approval was obtained, adding credibility to the research.
Suggestions for Improvement:
Clarification of Variables: The distinction between coping strategies and their effects on different dimensions of somatization and depression could be more detailed. It would be helpful to explain further how each coping strategy influences both somatization and depressive symptoms.
Discussion Expansion: While the findings are discussed in relation to previous research, more could be said about practical implications. How can hospitals and management teams implement stress-reduction interventions based on this study?
Control for Confounding Variables: It would be beneficial to account for other confounding factors that could influence depression and somatization symptoms, such as work shifts, sleep quality, or individual personality traits.
-I suggest to use these references at your discussion and introduction "
1-El-Ashry, A.M., Elsayed, S.M., Ghoneam, M.A. et al. Compassion fatigue and stress related to cardiopulmonary resuscitation: a study of critical care nurses’ experiences. BMC Nurs 22, 482 (2023). https://doi.org/10.1186/s12912-023-01640-y
2-Compassion fatigue in palliative care: Exploring Its comprehensive impact on geriatric nursing well-being and care quality in end-of-life. DOI10.1016/j.gerinurse.2024.05.024

3-An Assessment of Pediatric Nurses Awareness and Perceived Knowledge of Autism Spectrum Disorders: A Gulf State Survey .DOI10.1155/2023/4815914

4-Effectiveness of Cognitive Behavioral Therapy (CBT) on Psychological Distress among Mothers of Children with Autism Spectrum Disorder: The Role of Problem-Solving Appraisal. DOI10.3390/bs14010046
Limitations and Generalizability: The study acknowledges its limitation of sampling from a specific region (Hefei). Future research should aim to include a more diverse sample to improve the generalizability of the findings.
Additional Research Recommendations:
The authors should consider a longitudinal study to explore the long-term effects of coping strategies and social support on somatization and depressive symptoms.
Further exploration of intervention-based studies targeting stress management in nursing would provide practical tools for reducing somatization symptoms in healthcare workers.

Experimental design

no comment

Validity of the findings

no comment

---

## Round 0.2 · accepted · Accept

After revisions, all reviewers agreed to publish the manuscript. I also reviewed the manuscript and found no obvious risks to publication. Therefore, I also approved the publication of this manuscript.

·

Basic reporting

Dear Authors,

I would like to inform you that, after reviewing the revisions, I consider the article ready for publication.

Thank you for your efforts in improving the manuscript and for your collaboration. We remain available for any further necessary steps.

Experimental design

Dear Authors,

I would like to inform you that, after reviewing the revisions, I consider the article ready for publication.

Thank you for your efforts in improving the manuscript and for your collaboration. We remain available for any further necessary steps.

Validity of the findings

Dear Authors,

I would like to inform you that, after reviewing the revisions, I consider the article ready for publication.

Thank you for your efforts in improving the manuscript and for your collaboration. We remain available for any further necessary steps.

Additional comments

Dear Authors,

I would like to inform you that, after reviewing the revisions, I consider the article ready for publication.

Thank you for your efforts in improving the manuscript and for your collaboration. We remain available for any further necessary steps.

·

Basic reporting

None

Experimental design

None

Validity of the findings

None

Additional comments

None